# 5 Years of bipolar disorder conversations on Reddit: Methods, key topics and future directions

Caden Poh[1], Cheyanne Head[2], John-Jose Nunez[1], Laura Lapadat[3], Emma Morton[4], Collaborative RESearch Team to Study Psychosocial Issues in Bipolar Disorder (CREST.BD)[1‡], Erin E. Michalak[1]*

**1** Department of Psychiatry, University of British Columbia, Vancouver, British Columbia, Canada, **2** Department of Medicine, University of British Columbia, Vancouver, British Columbia, Canada, **3** Department of Psychology, McGill University, Montreal, Quebec, Canada, **4** School of Psychological Sciences, Monash University, Clayton, Victoria, Australia

‡ Membership list can be found in the Acknowledgments section.
* erin.michalak@ubc.ca

## Abstract

Bipolar disorder (BD) is a prevalent mood disorder that can be associated with serious personal, societal, and economic costs. Growing attention is being paid to the importance of including the experiences of people living with BD in research and healthcare system advancement. However, there is still much to be learned about what people with BD are prioritizing, and where their needs are not yet being met in research and clinical settings. For the past five years, the Collaborative RESearch Team to study psychosocial issues in Bipolar Disorder (CREST.BD) has facilitated the world's largest online BD question-and-answer event, an "Ask Me Anything" (AMA) hosted via Reddit. This event allows people internationally to submit BD-related questions to be answered by expert panelists. Altogether, 159 panelists with diverse expertise in BD have participated, and more than 2,000 questions addressed. After five years, the Reddit AMAs have become a large repository of questions that represent the needs and interests of individuals with BD and their supporters. This study used topic modeling, a natural language processing (NLP) method, to algorithmically extract key topics from AMA queries collected over five years. Queries were extracted using a python script and the Reddit API; BERTopic was used for topic modelling. Topic models of 10, 20, and an unrestricted number of topics were run; the 20 topic model was chosen as it best balanced specificity and breadth. The most common topic was BD misdiagnosis/differential diagnoses. Other topics included coping with daily struggles; understanding hypomania and suicidality; medication and use of substances such as psilocybin and ketamine; and supporting loved ones with BD. Future research in these areas would be beneficial for individuals with BD and their loved ones, and patients with BD may benefit from clinicians addressing these topics. Ethical issues are also discussed.

**Data availability statement:** The Reddit "Ask Me Anything" events containing the analyzed responses are publicly available on Reddit; links are collated here: https://www.crestbd.ca/previous-amas/. To ensure Reddit users maintain control of their responses (i.e., can remove their comments if desired), the extracted data file used in this analysis has not been made available. Code for analyses is publicly available on github, with the specific version archived here: https://doi.org/10.5281/zenodo.15092633.

**Funding:** CREST.BD education outreach activities are support by Otsuka-Lundbeck unrestricted grants. The funders had no role in study design, data collection and analysis, decision to publish, or preparation of the manuscript.

**Competing interests:** The authors have declared that no competing interests exist.

## Introduction

Bipolar disorder (BD) is a mental health condition affecting 2.5% of the global population [1], characterized by distinct periods of elevated mood (i.e., mania or hypomania) and depressed mood that can profoundly alter thinking and behaviour [2]. These mood states can seriously impact ability to engage in typical activities of daily life; BD is linked to personal, societal, and economic costs, and is estimated by the World Health Organization to be the 6th leading cause of disability worldwide [3]. Bipolar disorder is associated with higher mortality and shorter lifespan, due to concomitant physical health problems and incidence of suicide [4]. Despite the serious impacts of mental health conditions like BD, allocation of research funding towards the condition from both public and non-profit sources is disproportionate to the level of need, especially when compared with the amount of funding provided for physical health conditions [4,5]. Even among mental health conditions, funds are especially limited for BD–a report commissioned by Bipolar UK found that though BD accounts for 17% of the burden of mental illness in the UK, the condition accounts for only 1.5% of mental health research funding [4]. This underfunding has implications for research and service delivery. For example, the Bipolar UK report stated that 75% of women living with BD experience menstruation related symptom fluctuations, yet only 14% received information from their clinicians on hormonal influences in BD [4]. Discrepancies like this highlight how underfunding of BD might maintain a dearth of literature on clinically relevant subjects that could improve the health, wellbeing, and social functioning of individuals living with BD.

Given current inadequate funding of BD research, effective use of limited resources is vital. It is therefore necessary to take a considered approach towards selecting research directions most likely to yield impactful, clinically relevant outcomes. It is increasingly recognized that involving people with lived experience in shaping research agendas can enhance the relevance of research questions, as well as bolster community willingness to participate in research and increase the likelihood of research being translated into practice [6,7]. However, multiple studies have shown that people with lived experience of health conditions and researchers/clinicians do not always concur on priorities for health research [8–10]. For example, a priority-setting participatory research project found that the priorities of mental health providers were distinct from mental health service users: while providers elevated the importance of biomedical research, mental health service users showed more interest in social and psychological research [9]. A scoping review of 74 mental health priority setting exercises found that this pattern was reflected across studies, with rank ordering of research priorities differing between researchers and mental health service users [10]. Though both groups emphasized the primary importance of researching psychological interventions, researchers placed more emphasis on accurate and timely diagnoses, research methodologies, and research standards, while service users valued research concerning education and psychosocial aspects of mental health disorders. In recognition of such findings, many funders now ask that researchers demonstrate how their research plans

address community priorities [11]. Amplifying the voices of people with lived experience is also a key step towards addressing historical power imbalances and inequalities [12,13], as people with mental health conditions have often been excluded from making decisions about the healthcare and services which impact their lives.

To address the need for clinically relevant research and service provision, a number of consensus-building studies have been conducted to incorporate lived experience perspectives in developing research agendas and shaping service delivery for BD. Though these represent valuable contributions towards the goal of amplifying patient priorities, a recently submitted scoping review of existing consensus building approaches in BD identified several major limitations [14]. Notably, there was a lack of inclusion of stakeholders from diverse backgrounds: few studies were adequately diverse across age, race and ethnicity, gender and sexuality, spoken language, and history of engagement with mental health supports [14]. This low representation is likely due to small sample sizes and use of convenience sampling approaches [15], which bias participation towards individuals already well-engaged in research and healthcare networks. A second limitation identified by this scoping review was that mental health services users who are retained as participants in consensus-building projects over time tend to be those who are engaged and interested in the topic [14]. Repeated consultation of the same experts may bias findings, as those with more severe course of health problems [16], those facing economic barriers to participation [17], those with lesser interest in the research topics [18], and those from marginalized backgrounds may be less able to engage [19]. Altogether, there is a need to complement existing priority setting exercises with methods that can describe the unmet needs and interests of a broad spectrum of people with BD, especially those outside usual circles of participation.

One efficient, cost-effective, data-driven approach to understanding lived experience research priorities is to turn to online platforms that organically facilitate this dialogue. Individuals with mental health diagnoses increasingly utilize social media platforms to exchange and request information and support. A commonly used platform is Reddit, a social media service combined of "subreddits," topical communities where anonymous users discuss specific areas of interest [20]. Reddit has been shown to facilitate discussion on stigmatized topics, including mental health, likely due to its anonymity, group rules, broad accessibility, and power of community [21]; de Choudhury et al. found Reddit appears to facilitate greater disclosure and support about mental health than other platforms [22]. For example, the r/bipolar subreddit provides a social support network where users living with BD seek support, as well as share advice, information, coping strategies, and personal experiences [23]. Analyzing the experiences of individuals expressed on Reddit may give insight into the unmet needs and interests of people with BD, especially those with marginalized backgrounds, those not engaged in treatment, and those from rural areas. They may also capture perspectives of individuals for whom English is not a second language due to the availability of translation software. A semantic network analysis of r/bipolar found that common discussion topics were medication, financial problems, and sleep disorders [24]. Similarly, a topic modelling and sentiment analysis of r/bipolar found that users commonly share personal experiences, seek advice, and express emotions [23]. Studies like these demonstrate how the ability to quickly and automatically analyze this discourse using Machine Learning (ML) and Natural Language Processing (NLP) presents compelling opportunities to highlight public health research questions as expressed by individuals with lived experience in naturalistic contexts [25].

For the past five years, The Collaborative RESearch Team to study psychosocial issues in Bipolar Disorder (CREST.BD) has facilitated a 48-hour long "Ask Me Anything" (AMA) discussion via Reddit. CREST.BD is an international BD research network that has led knowledge exchange initiatives to disseminate BD research findings to the broader community. The AMA is hosted annually on World Bipolar Day to connect the public with a panel of BD specialists, including researchers, mental health clinicians, and people with lived experience of BD, many of whom are associated with CREST.BD as 'peer researchers'. The panelists answer user-submitted questions in a bidirectional interaction between knowledge users (Reddit users) and knowledge generators (here, academic researchers, clinicians, and peer researchers). The resulting knowledge exchange is thus personalized to the needs and interests of individual knowledge users who ask questions. This contrasts with other community-facing knowledge initiatives (e.g., webinars, podcasts, blogs), where information relay is based upon knowledge generators' impressions of what is relevant to the community, and the format is less responsive

to the needs of individual users. After five years of the AMA Reddit event, the AMA threads contain valuable insights into the priorities of BD communities and present an opportunity to highlight the important questions, topics of interest, and ongoing concerns of those affected by the condition.

The aims of this paper are twofold: 1) to provide an in-depth description of the methods we iteratively refined over the course of conducting five BD-specific Reddit AMA events, and; 2) to share the specific community priorities identified based on our prior BD AMAs using the topic modeling technique BERTopic. Our first aim will contextualize our analysis and enable other health researchers to benefit from our learnings from this novel form of knowledge exchange. Our second aim will provide new knowledge to the BD research community about the priorities of people living with BD around the world. Our hope is that this analysis will help guide future mental health research and knowledge exchange initiatives, whilst also amplifying the voice, questions, and concerns of people living with BD.

## Methods

### Procedure

**Reddit.** All AMA sessions were conducted on the r/IAmA subreddit, a Reddit-based community platform facilitating topic-oriented interviews (i.e., AMAs), where users from the general public pose questions to panelists. Per the rules of r/IAmA, each panelist provided proof of their identity by sharing a photo of themselves specifying the date, their name, and the title of the AMA event (e.g., see https://www.crestbd.ca/world-bipolar-day-ama-2023/). AMAs are public-facing in nature, allowing viewing of questions and responses, though only registered Reddit users may submit questions. The AMAs began each year on March 30th, World Bipolar Day, as a part of global awareness efforts for BD, and lasted 48 hours.

**Panelists.** The first World Bipolar Day AMA was paneled by CREST.BD director and co-director Dr. Erin Michalak and Dr. Steven Barnes in 2018. In response to demonstrated community interest, subsequent AMAs recruited larger numbers of panelists. A total of 159 expert panelists (see Supplementary File 1) were involved in the five AMAs occurring from 2019-2023. Criteria for panelists included: 1) lived experience of BD, and/or 2) academic/clinical expertise in mood disorders. Notably, a subset of AMA panelists contributed combined expertise, i.e., both lived experience or and academic/clinical expertise in BD. Consistent with CREST.BD's commitment to the principles of Community-Based Participatory Research (CBPR) [26], panelists were selected with expertise in areas corresponding to topics of interest raised in prior AMAs. A proportion of selected panelists (63%) were existing members of the CREST.BD network, while others joined from other BD or mood disorder research and support networks, such as the International Society for Bipolar Disorders (ISBD), Bipolar UK, and Depression and Bipolar Support Alliance (DBSA). Panelists were also located in different geographies around the world, with representatives from most continents. Overall, the approach to panelist recruitment ensured that the AMA discourse included a diverse spectrum of perspectives and expertise, in alignment with CREST.BD's dedication to inclusivity and collaborative inquiry.

**Promotion.** The AMA was promoted through CREST.BD knowledge exchange platforms, including the CREST.BD Bipolar Blog, social media channels, and a mailing list. The AMA was also promoted by community partners, e.g. through social media postings and word of mouth.

**Facilitating panelist contributions.** Panelists answered questions via a shared Google document collating user-submitted questions. Considering the large number of panelists, varying levels of familiarity with Reddit, and a need to avoid multiple users answering the same question simultaneously, it was more feasible to coordinate activities in a shared document than have panelists respond directly via Reddit. Google Drive was chosen due to its collaborative functionality and worldwide accessibility, which was essential considering panelists were internationally located. All panelists received access to the Google document in advance, and were provided with video instructions on how to use it. A single Reddit account (u/crest_bd) was created to represent all panelists engaged in the AMA session, which was operated by a smaller internal AMA team (2-4 research assistants, including co-authors CP and LL). After a Reddit user published a new post

to the AMA, the team would paste it into the document, collating user-submitted questions as separate table rows in the Google document. Questions were numbered and pasted in the order of submission. Panelists then wrote their answers below the questions, and signified when their answers were complete by writing the word "DONE" in capital letters. Internal AMA team members would then use the u/CREST_BD Reddit account to paste the answers as Reddit replies to the question posts.

### Ethical considerations

**Lived experience panelists.**  Due to the sensitive nature of topics discussed during the AMA, a protocol was implemented to safeguard the well-being of panelists, particularly those living with BD. Participation as a panelist was voluntary, with prospective panelists given prior access to detailed information about previous AMAs and example questions to facilitate informed decision-making prior to their participation. Panelists were also under no obligation regarding time commitments or the length of their responses, affording them the autonomy to engage selectively with questions that were within their comfort levels. To enhance support mechanisms and as a proactive risk management strategy, a dedicated WhatsApp group was established to connect panelists and foster support when needed. Furthermore, members of the CREST.BD research and clinical team led by EEM, acting in her capacity as director, maintained continual availability throughout the AMA periods via email and WhatsApp to ensure that panelists encountering challenges received timely support.

**Reddit data usage.**  Due to the public nature of Reddit content, as well as its (relative) anonymity, researchers commonly consider data from Reddit to be open-use and to therefore not require ethical board permission prior to use [27]. Additionally, traditional ethics consent procedures are challenging, if not impossible, to implement en masse across large groups of online users [27]. Due to the AMAs containing thousands of responses, and because AMA content was not originally intended to be used for a research study, implementing formal consent procedures was unfeasible. We therefore did not seek formal ethical approval for the study presented here. However, as prior researchers have emphasized, when working with online content produced by at-risk populations, such as those with stigmatized identities, it is especially important to consider potential ethical concerns [27]. Indeed, considering the high prevalence of data privacy concerns [28], analyzing any form of online data may be a source of discomfort to the target population.

Several layers of confidentiality, as well as points of possible vulnerability, were considered. In terms of confidentiality, Reddit users can adopt any username they want, and typically choose anonymous usernames. Though our CREST.BD events are focused on the needs of individuals with BD, r/IAmA is not a BD-specific subreddit, and users did not need to identify any personal information, such as having a diagnosis, to partake. Additionally, given the Q&A structure of the event, users tend to frame their responses as brief queries, rather than personal stories. Importantly, the data analysis, described below, analyzes responses in aggregate: only four researchers viewed the quotations from participants during analysis (co-authors CH, JJN, LL, and EEM), which was considered the minimum number of researchers possible to effectively perform this analysis. However, it is always possible for any Reddit user to include identifiable information in their posts or user profiles. Also, although Reddit posts are public-facing, users can always choose to delete their posts from Reddit, but they cannot do this with published data sharing their responses. Further, though it is unclear whether a particular user lives with BD based on their choice to post to the AMA, it is implied that many users either live with BD or are a carer or loved one of someone who does. To maximize confidentiality, all usernames were omitted from the analysis and data sharing processes. We have also abstained from using direct quotes from users, as these may be indexed in search engines [29]. We instead present our results in aggregate and use paraphrasing of quotes. Between the anonymity of Reddit, and our additional measures taken to ensure confidentiality, we believe the identity of AMA participants has been protected to the best of our ability.

## Topic modeling

To understand the topics of questions posed by AMA participants, topic modeling was used employing the BERTopic language modelling technique. Co-author JJN first extracted all participant posts from across five years' of AMAs automatically using Python. Only top-level posts were included (i.e., comments that were posted directly in response to the AMA, as opposed to "replies" that are posted in response to a particular post). We excluded all posts by panelists and all replies. As some users still posted new top-level posts meant as replies, a team member (CH) manually filtered remaining posts that appeared to be replies and/or did not have any substantive content, e.g. "Thank you Tom!" A second team member (LL) double-checked all posts to be excluded.

After filtering the final set of participants posts, we conducted topic modeling using BERTopic [30], a modern analysis technique that has been used previously in studies of mental health discourse on social media [31–33]. Topic modeling is an application of NLP, the field of artificial intelligence that uses machine learning to interpret, summarize, and produce human language. Topic modeling extracts a set of topics that describe subsets of units of information, or "documents", in a collection. In our case, these "documents" are individual Reddit user posts on the AMAs. To accomplish topic modeling, BERTopic first transforms each document into a numerical vector, which can be thought of as a coordinate in space. Clusters of vectors indicate relatedness between their contents, and thus represent topics detected by BERTopic across the aggregated documents. BERTopic is then able to convert these numerical representations back into human language to describe the topics, yielding lists of terms and phrases that capture the essence of each topic.

BERTopic accomplishes these steps in a modular fashion, with different techniques swappable for each constituent step. For our deployment, we used default choices similar to those used in a recent study [31], including UMAP [34], HDB-Scan [35], and CountVectorizer from scikit-learn [36]. As the number of topics can influence their granularity and show different patterns, we used the *nr_topics* hyperparameter to run the analysis with a different number of topics set. The *nr_topics* setting determines a priori how many topics BERTopic will extract from the data. The technique was first run with no topic limit set (the default setting) and then with 10 and 20 topic limits. The final analysis chosen consisted of 20 topics, which was determined by researcher consensus to best balance topic specificity with interpretability.

To represent the final topics, we used the default Maximal Marginal Relevance representation KeyBERT80, which was used to extract keywords, and OpenAI's ChatGPT 3.5 Turbo model81. The ChatGPT 3.5 Turbo model is a computational model that can interpret and generate human language, or large language model (LLM), and was employed to interpret the topic representations generated by BERTopic and assign short labels representing each topic.

Further details on our implementation, including all code and specific settings, are publicly available through in a Github repository (https://github.com/jjnunez11/world_bipolar_day_ama_bertopic), with the specific version used in this work archived on Zenodo (https://doi.org/10.5281/zenodo.15092633). We are not able to include the raw posts from Reddit to preserve the privacy of participants.

## Results

### Reach and engagement

The AMA events between 2019 and 2023 (Table 1) generated substantial international participation and engagement, and, to our knowledge, represent the largest online BD-specific AMAs to date. For example, the 2023 event included 68 panelists, generated almost 1,700 comments, and exceeded 1.1 million views on Reddit. Cumulatively, the five events garnered 11,346 upvotes and 6,159 comments. In terms of panelist engagement, in total 159 BD experts from 14 countries participated on one or more occasions, reflecting broad international representation.

**Table 1**. Statistics for each year's AMA.

| AMA | Total views* | Total shares* | Upvotes | Comments | Panelists | Questions | Panelist Responses |
|---|---|---|---|---|---|---|---|
| 2023 | 1.4 million | 1,500 | 1,695 | 1,575 | 68 | 534 | 603 |
| 2022 | 1.4 million | 960 | 5,147 | 2,250 | 44 | 636 | 547 |
| 2021 | 2.8 million | 2,600 | 809 | 1,035 | 28 | 341 | 291 |
| 2020 | Not available | Not available | 3,438 | 1,078 | 14 | 408 | 272 |
| 2019 | Not available | Not available | 257 | 221 | 5 | 94 | 78 |
| Total | -- | -- | 11,346 | 6,159 | 159 | 2,013 | 1,791 |

*Note: This data is only available for years 2021, 2022, and 2023, as Reddit's Post Insights feature was released in 2021.

## Topic modeling

In the first topic model, which was run using the described hyperparameters but without setting a topic limit, BERTopic found 45 topics, covering 1,539 of 3,017 participant posts (51.0%). We then used *nr_topics* to constrict the number of topics to 10 and 20, covering the same number of posts.

In Table 2, we show the result of using topic representation generated by ChatGPT 3.5 Turbo with *nr_topics* set to show 20 topics, which we found to be the best balance between specificity of topics without needless granularity. The resultant 20 topics cover a wide range of BD-related themes, with the common topic, Topic 1, being interpreted by ChatGPT 3.5 Turbo as "Misdiagnosis of BD." This arose 575 times across the 1,539 documents included in the analysis. This topic encapsulated questions about misdiagnosis and differential diagnosis, and open-ended questions and critiques about the diagnostic criteria for BD. The next most common topic, Topic 2, relates to the AMA itself, with users expressing gratitude, commenting (mainly positively) about the quality of the answers, and, in some cases, commenting on whether questions had been answered and musing about preferential responses to some questions over others. Topic 10 clustered posts similar to Topic 2 without clear differences aside from the comments being generally briefer and less specific. Topics 3, 14, and 17, and topics 5, 7, 11, and 13 were related to substance use) and medications respectively. Users expressed interest in novel pharmacotherapies, especially those using substances commonly used for recreational purposes, like cannabis, psilocybin, and ketamine. Regarding prescription drugs, user comments varied from questions about their current medications (e.g., asking about the therapeutic dose of lithium, or whether an antipsychotic dose may be "too high"); querying about pregnancy and medication safety; asking how to minimize unwanted side effects such as weight gain and fatigue; and sharing both positive and negative testimonials about their use of specific medications. Topic 18 addressed both use of DBT workbooks as well as soliciting panelists' opinions about Thomas Szasz's book, *The Myth of Mental Illness*; the documents in this topic seem to have been clustered together based on the shared feature of asking about mental health books.

Other topics were related to BD symptomatology, and ways for coping with one's own BD or ways of supporting loved ones with BD. Three topics focused on diagnosis and symptoms: Topic 15, "Cyclothymia" "Suicidality"; and Topic 6, "Understanding Hypomania". When asking about these topics, some users were requesting diagnostic definitions, or information regarding prevalence rates. Others were seeking to understand their own experiences and stated their mental health symptoms (e.g., explaining anxiety disorder symptoms and questioning whether that fits with a diagnosis of cyclothymia). Topic 8, "Coping with Daily Challenges", included questions about mood tracking and recognizing an episode may be imminent, and how to cope with challenges like BD-related interpersonal problems and lowered quality of life. Another topic, Topic 16, was found specifically for the ketogenic diet), with users asking about the present research on the topic, conversing about media suggesting this diet may be supportive of BD, and sharing their own experiences (with some recommending the diet, and others advising against it). BERTopic also identified a cluster of posts about how to support family members with BD, Topic 4, which often focused on how to encourage a friend, family member, or partner to seek appropriate treatment for diagnosed BD, or to convince a loved one suspected of having BD to seek a diagnosis.

**Table 2**. Final topics, keywords, and representative quotes per topic.

| Topic | Participants Posts in Topic | OpenAI Topic Representation[a] | Top 10 Words in Topic[b] | Representative Paraphrased Quote[c] |
|---|---|---|---|---|
| 1 | 575 | ['Bipolar Disorder Misdiagnosis' | 'bipolar', 'disorder', 'bipolar disorder', 'im', 'diagnosed', 'people', 'just', 'adhd', 'like', 'know' | "Are there any changes to the bipolar disorder diagnosis that you expect or hope to see in the next edition of the DSM?" |
| 2 | 157 | 'Expressing gratitude and struggles' | 'thank', 'thanks', 'people', 'thats', 'im', 'like', 'dont', 'time', 'just', 'answer' | "Thank you for sharing your story! Living with these symptoms is difficult and it gives me hope and makes me happy to hear someone getting through it. I'm proud of you!" |
| 3 | 130 | 'Marijuana and Bipolar Disorder' | 'marijuana', 'use', 'bipolar', 'ketamine', 'cannabis', 'thc', 'weed', 'treatment', 'psychedelics', 'psilocybin' | "Yeah, like you, I've taken THC edibles that messed up my damn mind. I puked so much I was crying and my eyes were bloodshot and I needed to go to the ER. I used to be able to use a low-percent CBD oil, but now that I'm medicated I can't tolerate any kind of weed." |
| 4 | 91 | 'Supporting Bipolar Family Member' | 'help', 'family', 'person', 'bipolar', 'friend', 'doesnt', 'support', 'time', 'know', 'hes' | "I'm desperate to know what to do. My loved one refuses any kind of help or medication even though there's lots available. What can I do? Who can I reach out to? I don't know what to do with someone in this state. How can I get him help when he refuses everything? There must be something I can do. Please, any help or advice is appreciated." |
| 5 | 85 | 'Lithium therapeutic index monitoring' | 'lithium', 'effects', 'years', 'kidney', 'levels', 'effective', 'ive', 'like', 'therapeutic', 'bipolar' | "Why are psychiatrists so convinced that the therapeutic window for lithium requires a moderate dose? I've read articles about low levels of lithium helping brain health and lower levels of violence when the drinking water has naturally higher lithium levels. Is there any research on lithium microdosing for bipolar disorder? Can more research be done on this?" |
| 6 | 76 | 'Understanding Hypomania' | 'hypomania', 'mania', 'hypomanic', 'episode', 'episodes', 'manic', 'like', 'brain', 'dont', 'know' | "Is it possible for people with BD to have SOME but not all symptoms of hypomania? Even not appearing manic at all? I think I've had this. I've found some information about subthreshold and subsyndromal hypomanic symptoms, but I didn't really understand it or the terminology." |
| 7 | 73 | 'Mood effects of BD treatment' | 'bd', 'diagnosed bd', 'diagnosed', 'mood', 'like', 'does', 'thanks', 'im', 'bpd', 'ive' | "I feel like with lamotrigine, my depressions aren't deep and dark like before, and I've also been able to avoid hypomanic episodes by living in a more consistent way. I've done CBT too. But even though I'm not having episodes, I still have trouble feeling emotionally consistent. Are these mini mood episodes that are just not as severe as usual now that I'm treated? What kinds of treatments can help with this?" |
| 8 | 68 | 'Coping with Daily Challenges' | 'im', 'just', 'dont', 'like', 'dont know', 'thats', 'know', 'really', 'day', 'worse' | "Omg, when manic I crave winning. It helps if my husband gives me some sort of 'win.' For example, 'Let's not remodel the room this week, but we can go to the store and take pictures of some options.' " |
| 9 | 49 | 'Bipolar Sleep Cycling Disorder' | 'sleep', 'cycling', 'rapid', 'rapid cycling', 'circadian', 'mixed', 'days', 'bipolar', 'rhythm', 'light' | "Wanted to share an unusual treatment that's worked for me. I've had a dx of BD for a long time now, but for the first ten years my sleep was HORRIBLE. At its worst I barely slept when manic and slept 18 hours a day when I was depressed. My psychiatrist gave me stimulant medication, like for ADHD. Because I take it at the same time every morning, I feel energetic at the same times and get tired at the same time. I have a regular circadian rhythm and have been significantly less symptomatic since." |
| 10 | 31 | 'Gratitude and Hope' | 'thank', 'hope', 'sorry', 'reply', 'thank reply', 'im sorry', 'im', 'really helpful', 'response', 'better' | "Thank you for taking the time to speak with us. I've read a lot about this topic, and it's lovely to speak with actual researchers!" |

*(continued)*

| Topic | Participants Posts in Topic | OpenAI Topic Representation[a] | Top 10 Words in Topic[b] | Representative Paraphrased Quote[c] |
|---|---|---|---|---|
| 11 | 29 | 'Managing Sleep with Seroquel' | 'seroquel', 'sleep', 'mg', 'dosage', 'combo', 'taking', 'drooling', 'helps sleep', 'helps', 'got' | "Me too, I take lamictal in the day and seroquel at night. This combo really helps which tbh surprised me." |
| 12 | 28 | 'Depictions of Bipolar Disorder' | 'bipolar', 'tv', 'ben', 'movie', 'accurate', 'silver', 'ozark', 'seen', 'character', 'disorder' | "What are some portrayals of bipolar disorder in popular media that are especially good or bad? Some that I think of are Silver Linings Playbook, Next to Normal, and one arc in the 2000s Degrassi." |
| 13 | 28 | 'Lamictal Lamotrigine Interactions' | 'lamictal', 'lamotrigine', 'rash', 'effexor', 'effects', 'taking lamictal', 'just', 'taking', 'does lamotrigine', 'think effexor' | "I tried lamictal and got the rash within a day after titrating from 25 to 50mg. Ofc I had to stop taking it immediately. What I want to know is if I could try again with slower titration? Is it possible it would work then, or is it like an allergy where no matter what lamictal is poison to me?" |
| 14 | 25 | 'Bipolar Pregnancy Medication Safety' | 'pregnancy', 'baby', 'medication', 'medications', 'bipolar', 'lamictal', 'concentrations', 'safe', 'stay', 'mood' | My husband and I decided to have bio kids, but it took years of agonizing about whether or not this would be the right choice. We pay attention to mood their changes and we're teaching them DBT skills so they can learn to manage emotions. My hope is that if they do end up having BD, we'll all be ready to handle it." |
| 15 | 23 | 'Understanding Cyclothymia' | 'cyclothymia', 'anxiety', 'panic', 'cyclothymic', 'disorder', 'drinking', 'depression', 'social anxiety', 'atypical depression', 'bipolar' | "Do you think that cyclothymia is recognized less than it should be? Why is there so little information about cyclothymia and why is BD mainly talked about in terms of bipolar I and II?" |
| 16 | 16 | 'Keto Diet and Bipolar Disorder' | 'keto', 'diet', 'palmers', 'keto diet', 'dr', 'metabolic', 'relief', 'mental disorders', 'research', 'disorders' | "Are there any studies showing that the keto diet can help improve mood and symptoms in bipolar disorder?" |
| 17 | 15 | 'Alcohol Use in Bipolar Disorder' | 'alcohol', 'drink', 'alcoholism', 'bipolar', 'drinking', 'taking', 'medication', 'alcohol use', 'days really', 'blackout' | "Is it more common for people with BD to have issues with alcohol and drugs? Alcohol was my only way to cope with BD as a teenager. My parents wouldn't let me go to therapy and I would drink until I blacked out to cope." |
| 18 | 15 | 'Thomas Szasz Book Review' | 'book', 'read', 'thomas', 'books', 'dbt', 'read book', 'lately', 'ive', 'psychiatrist', 'finally' | "I've been in a mixed hypomanic/depressive episode for months and it's been destructive for my work and relationships. I'm using the DBT workbook with my therapist and it's great. I also appreciate that it has some humour and snark to it, because workbooks can be a drag." |
| 19 | 14 | 'Bipolar Suicide Rates' | 'suicide', 'life expectancy', 'expectancy', 'commit suicide', 'rate', 'higher', 'commit', 'mental illness', 'rates', 'studies' | "Sorry if this is a difficult, but I'd like to know: What are the stats on suicide and bipolar? I want to clear up confusion and misinformation considering how stigmatized this subject is. I've heard some stats, like how women survive attempts more than men, but nothing bipolar disorder specific." |
| 20 | 11 | 'Managing Bipolar Disorder Diagnosis' | 'bp2', 'bpii', 'bp', 'bp1', 'bp2 diagnosis', 'aging', 'life', 'reached', 'struggled', 'meds' | "Is it possible to have had a diagnosis of BD II for 30 years, and then be considered BD I?" |

Topics resulting from using BERTopic on 1,539 participant posts.

[a]Representation of a topic generated by OpenAI's ChatGPT 3.5 Turbo Model.

[b]Default ten-word representation generated by BERTopic.

[c]Participant post categorized by BERTopic within this topic and selected by EEM, paraphrased to preserve participant anonymity.

## Discussion

The goals of this paper were to: 1) describe the methods used by CREST.BD to deliver five consecutive years of a BD-specific Reddit AMA, and 2) to apply NLP techniques to extract key themes from these conversations that signpost the research priorities of people living with BD. The AMA events received significant community engagement, with 11,346 upvotes, 6,159 comments, and 2,013 questions submitted in total. This indicates community need and interest, and demonstrates the utility of online community events for knowledge exchange and consensus building activities. The event also attracted the efforts of panelists with varied BD-related expertise internationally, with 159 panelists in total. In describing the methods to bring about this successful event, our hope is that other research teams will take on similar events and improve accessibility of knowledge mobilization and exchange with populations living with chronic health conditions.

The present analysis highlights the potential for employing Reddit-based community engagement events to gain insights into the unmet needs of people with mental health conditions. A previous scoping review highlighted that the largest priority setting exercise for BD to date was a survey with $n$=198 participants [37]. This survey was also quantitative in nature, meaning that initial priorities for inclusion in the survey were generated by the research team, and may not have reflected all topics of concern or interest to people with BD. Although we did not calculate the number of unique respondents (and indeed, this is not feasible, given the potential of users holding multiple Reddit accounts), the number of likely participants in the AMA far exceeds the sample size participating in any BD priority-setting or consensus building exercise to date. Online community events like these AMAs may therefore be a new avenue to gather a broad range of community perspectives to inform future priority setting and consensus building exercises. For example, a first step in Delphi study is to generate items for consensus ranking; these items are typically generated by a research team from a review of the literature, prior qualitative work, or responses from the Delphi panel to open-ended survey items in the first round of data collection. By leveraging online community events, it would be possible to generate items for inclusion based on a much larger sample than is typical for Delphi panels [38]. Further, the community-oriented, bidirectional nature of the knowledge exchange provides an extensive resource for community members who participate, as well as those who prefer to browse without directly engaging, helping to fill critical knowledge gaps affecting under-served populations.

Notably, use of online community events to help inform priority setting and consensus building addresses a recognized challenge in this area: that not all individuals are invited or able to participate in patient engagement activities. Numerous barriers to engagement persist, including transportation and inadequate time to participate [19,39], as well as personal factors such as worries about embarrassment, privacy concerns, and lack of self-confidence [19]. These barriers disproportionately impact people of marginalized backgrounds and identities [38], which is especially problematic considering these individuals may have differing, and often more negative, experiences of healthcare systems that are crucial to capture in priority-setting and consensus-building activities [19]. Challenges of living with BD may themselves impact engagement, as serious mood symptoms, comorbidities, or associated cognitive and memory difficulties may inhibit participation. Further, individuals who challenge dominant views in healthcare may be less likely to be included or listened to during priority setting studies [18,40], suggesting that perspectives that do not align with prevailing research priorities may be missed. Online events like AMAs can help address these traditional barriers to entry: they are accessible globally to anyone with internet access, they are anonymous and require no transport, and they are asynchronous and require little time commitment from participants.

Although analyses of social media data do not substitute for direct engagement with communities, analyses of such data may assist in highlighting needs and perspectives previously unrepresented in research of patient priorities. In the present study, the topics identified by BERTopic displayed several differences between the primary concerns of AMA users versus the outcome of nine earlier priority-setting and consensus-building studies in BD. For example, the most prominent topic within the AMA questions was BD misdiagnosis. This can be contrasted to previous consensus-building

studies, few of which identified needs related to diagnosis, commonly focusing on issues related to treatment (e.g., medications and their side effects) instead [14,37,41–43]. This discrepancy may be due to the convenience or purposive sampling techniques employed by patient engagement projects, which may not include individuals with unclear or ambiguous diagnoses, or because participants may be uncomfortable discussing these questions if it has been established that a specific diagnosis is required for participation. Community interest in substance use was represented across several of the NLP-identified topics, with particular interest of AMA participants in the risk and benefits of often-recreationally substances like cannabis and psychedelics. Though such substances are known to be used in naturalistic settings by people living with BD [44], their therapeutic potential was not identified as a key topic in earlier consensus building studies. Such topics may be difficult to discuss in the context of traditional patient-engaged settings, where participants may lack anonymity and fear repercussions or stigmatization (especially when discussing illicit substances). Another key divergence from previous research is the prominence of caregiver concerns: in this analysis, questions about supporting loved ones was the fourth most discussed topic, yet only one previous consensus building study focused on the information needs of caregivers [45]. This underlines another strength of the online community event approach: because participation is open to any users, populations who are typically not considered by researchers may access the event and partake, as opposed to conventional approaches where the population of interest must be specified a priori. We also observed concordance between some topics of prior priority setting studies and the AMA topics of interest, such as sleep, pharmacological management of BD, and lifestyle influences and effective coping studies. Altogether, the NLP analysis identified several topics undiscussed in previous research seeking to identify priorities of patients with BD, highlighting a need to more carefully attend to issues of diversity and retention (as discussed by O'Donnell et al.) [14] to broaden the relevance of consensus building research.

### Ethics

A strength of this study is its considered approach to ethics, which are not always taken into account in social media research [27,46]. When assessing the ethics of the study, we reflected upon a) our stance in the research process and b) how our findings may serve the community. This stance differs from how Reddit research is commonly conducted. Typically, prior research has involved researchers collecting volumes of data from existing subreddits that the researchers have no previous involvement with [27]. This process may be viewed by users as exploitative; for example, an analysis of Reddit resource reports found that many Reddit users deleted their publicly available comments following their inclusion in research articles, suggesting users may have been uncomfortable with their data being used this way [29]. Unlike such studies, which analyze extant Reddit data, this AMA project was not intended at the outset to be used for research purposes; the primary aim of the events was to increase public access to BD research and treatment evidence through a broadly accessible platform. However, due the disparity in health data that matches service user priorities, and the concomitant impetus to generate research relevant to populations of interest, a post hoc decision was made after running five years of events to share knowledge from these discourses with research and lived experience communities. Analyzing AMA data was a low-cost, community-engaged way to identify key areas of need prioritized by people with BD and their loved ones. Findings can be conveyed to the BD research community through the present paper, and distributed amongst researchers of diverse BD expertise in the CREST.BD network. To ensure privacy of participants, all quotes have been paraphrased. Further, results of this study, and any research projects based upon it, can be disseminated back to the Reddit AMA community through later AMAs and other CREST.BD knowledge exchange platforms (e.g., blogs, social media, our talkBD podcast). Altogether, the motivation to use the data for these analyses was our assessment that the questions reflect community need, and represent an opportunity to translate concerns relevant to the BD community into academic research goals. We believe the potential for these findings to benefit individuals with BD is substantial.

## Limitations

Though Reddit holds many strengths, such as its wide accessibility and low barrier to entry, there are also limitations to the platform. The platform is primarily in English, and most AMA questions were submitted in English. Reddit users are themselves a specific population, estimated to skew younger, male-identified, North American, and more highly educated than the general population [47] (although demographics vary by subreddit) [48]. Members have self-selected to be Reddit users and dedicate time to use of this platform as opposed to other platforms, and may not be representative of the full population. It was not possible to collect demographic information about AMA participants, so though we consider it very likely that we have reached populations typically less engaged in mental health research, such as people who choose not to or cannot access mental health support, people from diverse ethnic and socioeconomic backgrounds, and people from varied geographical areas, this cannot be stated with certainty. Additionally, participants in the AMA did not have a confirmed diagnosis. Thus, it cannot be definitively stated the questions analysed using BERTopic were submitted by individuals with BD.

In terms of our data analytic approach, BERTopic itself has some limitations. Firstly, it requires some researcher subjectivity in its implementation. For example, hyperparameters set by the researcher, such as the number of topics and clustering settings, influence the format and content of results. Further, we used a live-updated, closed-source LLM to label the topics; as this LLM is constantly updated, different results may be attained if the analyses were run at a later point. Thirdly, though our implementation of BERTopic with ChatGPT 3.5 Turbo identified meaningful categories, the conceptual relationships this technique makes between documents can differ from human-like judgment, e.g. making connections the basis of trivial components of the text or applying a label that does not accurately capture the documents. In our analysis, the majority of topics were meaningful and were considered by the research team to be strongly conceptually related. However, one of the 20 topics was labelled "Thomas Szasz book review," but the contents of this topic revealed a cluster of posts about mental health books, mainly Szasz's book and DBT workbooks; the label applied was not appropriately descriptive of the topic contents. Thus, it is important to underline that computational techniques like this are not strictly objective and require researcher interpretation. Altogether, this analysis [49] should not be interpreted as definitive, but rather as a series of promising results using a state-of-the-art technique configured to its typical settings. Other settings and techniques for such analyses could be explored in the future.

## Conclusion

Our results indicate that Reddit is a valuable and largely untapped resource for health researchers, and presents an opportunity for bidirectional knowledge generation and exchange between researchers and users. AMAs may be a new avenue for informing priority setting and consensus building exercises that engage a much larger sample of patients than is possible in real-world settings. Our analysis identified several more novel community priorities, including information on misdiagnosis and differential diagnosis; research into new pharmacological approaches, notably use of historically recreational drugs such as cannabis and psilocybin; studies examining diet and BD, especially the ketogenic diet; research into cyclothymia; and a need for knowledge and resources to support loved ones with BD. There was also concordance with previous priority-setting exercise in areas like information on sleep and BD, including effects of circadian rhythm disruptors like artificial light; methods for day-to-day coping; and a notable expressed need for information about pharmaceutical medications, such as appropriate dosage or usage when pregnant. Future research into these lived experience-informed priority areas holds potential to enhance the quality of care and clinical management of BD.

## Supporting information

**S1 File. Promotional blog with panelist proof photos.**
(JPG)

**S2 File. Promotional event graphic.**
(JPG)

## Acknowledgments

We would like to thank the Reddit users who have participated in this event over the past five years for sharing their questions and making this research possible. We would also like to thank the panelists who have contributed their time and expertise to the AMA over the past five years. The group CREST.BD is led by author Erin E. Michalak (erin.michalak@ubc.ca), and includes individual authors Caden Poh, John-Jose Nunez, Laura Lapadat, and Emma Morton. A full list of network members can be found at https://www.crestbd.ca/about/team.

## Author contributions

**Conceptualization:** Caden Poh, Cheyanne Head, John-Jose Nunez, Laura Lapadat, Emma Morton, Erin E. Michalak.

**Data curation:** Caden Poh, Cheyanne Head, John-Jose Nunez, Laura Lapadat.

**Formal analysis:** John-Jose Nunez.

**Funding acquisition:** Erin E. Michalak.

**Investigation:** Caden Poh, Laura Lapadat.

**Methodology:** Caden Poh, John-Jose Nunez, Laura Lapadat, Erin E. Michalak.

**Project administration:** Caden Poh, Cheyanne Head, Laura Lapadat, Erin E. Michalak.

**Resources:** Caden Poh, John-Jose Nunez.

**Software:** John-Jose Nunez.

**Supervision:** Erin E. Michalak.

**Visualization:** Caden Poh, Laura Lapadat.

**Writing – original draft:** Caden Poh, Cheyanne Head, John-Jose Nunez, Laura Lapadat, Emma Morton, Erin E. Michalak.

**Writing – review & editing:** Caden Poh, Laura Lapadat, Emma Morton, Erin E. Michalak.

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
