## [Decision Letter · Decision Letter 0]

12 Jun 2025

PONE-D-25-19143

5 years of bipolar disorder conversations on Reddit: Methods, key topics and future directions

PLOS ONE

Dear Dr. Poh,

Thank you for submitting your manuscript to PLOS ONE. After careful consideration, we feel that it has merit but does not fully meet PLOS ONE’s publication criteria as it currently stands. Therefore, we invite you to submit a revised version of the manuscript that addresses the points raised during the review process.

ACADEMIC EDITOR:

The reviewers have provisionally accepted this manuscript with Reviewer 2 having a very minor edit of adding "the" on line 455. Please revise the manuscript to include this edit and resubmit for acceptance.

We look forward to receiving your revised manuscript.

Kind regards,

Souparno Mitra, M.D.

Academic Editor

PLOS ONE

Journal Requirements:

2. Thank you for stating the following financial disclosure: [CREST.BD education outreach activities are support by Otsuka-Lundbeck unrestricted grants.]

3. One of the noted authors is a group. [The Collaborative RESearch Team to study psychosocial issues in Bipolar Disorder (CREST.BD)]. In addition to naming the author group, please list the individual authors and affiliations within this group in the acknowledgments section of your manuscript. Please also indicate clearly a lead author for this group along with a contact email address.

5. Please remove all personal information, ensure that the data shared are in accordance with participant consent, and re-upload a fully anonymized data set.

6. We note that [Supplementary File 2 - Blog with Proof Photos] includes an image of a participant in the study].

Reviewers' comments:

Reviewer's Responses to Questions

**Comments to the Author**

1. Is the manuscript technically sound, and do the data support the conclusions?

Reviewer #1: Yes

Reviewer #2: Yes

2. Has the statistical analysis been performed appropriately and rigorously?

Reviewer #1: I Don't Know

Reviewer #2: Yes

3. Have the authors made all data underlying the findings in their manuscript fully available?

Reviewer #1: No

Reviewer #2: Yes

4. Is the manuscript presented in an intelligible fashion and written in standard English?

Reviewer #1: Yes

Reviewer #2: Yes

5. Review Comments to the Author

Reviewer #1: This manuscript offers a highly original and timely contribution to mental health research by leveraging five years of Reddit "Ask Me Anything" (AMA) events hosted by CREST.BD to explore the lived experiences and priorities of individuals affected by bipolar disorder. A key strength is its innovative application of BERTopic modeling to a large and diverse dataset of over 3,000 Reddit questions, enabling the extraction of organically expressed themes from a broad, international audience. The study excels in its community-engaged design, involving panelists with both lived experience and clinical or research expertise, which enhances the relevance and ethical integrity of the work. It thoughtfully addresses privacy and data ethics, taking proactive measures such as paraphrasing user quotes and removing identifying information. The study also contributes rich thematic insights—such as community concerns around misdiagnosis, alternative treatments (e.g., psychedelics), and caregiver challenges—which are underrepresented in traditional research agendas. Furthermore, the open-access sharing of analysis code aligns with best practices in transparency and reproducibility. These strengths collectively underscore the study’s value as both a methodological guide and a data-driven resource for informing future research and healthcare policy in bipolar disorder.

Reviewer #2: Well written article : 5 years of bipolar disorder conversations on Reddit: Methods, key topics and future directions. I have included edits/suggestions on the attached PDF. The limitations were unique to reddit, which made it an even more interesting read.

6. PLOS authors have the option to publish the peer review history of their article (what does this mean?). If published, this will include your full peer review and any attached files.

Reviewer #1: No

Reviewer #2: **Yes:** Arun Prasad

---

## [Author Response · Author response to Decision Letter 1]

18 Nov 2025

Thank you for reviewing our manuscript. This rebuttal letter contains responses to each point.

Our manuscript has been styled to meet the PLOS ONE style requirements.

2. Thank you for stating the following financial disclosure: [CREST.BD education outreach activities are support by Otsuka-Lundbeck unrestricted grants.]

The funders had no role in the study, we now state in the amended cover letter that "The funders had no role in study design, data collection and analysis, decision to publish, or preparation of the manuscript.

3. One of the noted authors is a group. [The Collaborative RESearch Team to study psychosocial issues in Bipolar Disorder (CREST.BD)]. In addition to naming the author group, please list the individual authors and affiliations within this group in the acknowledgments section of your manuscript. Please also indicate clearly a lead author for this group along with a contact email address.

This has now been added to the acknowledgments section of the manuscript.

We thank the Editor for this comment. We took a considered approach to the ethics of this paper. As we explain in lines 221-226, for data of this type (large volumes of anonymous online data), implementing typical consent procedures was not determined to be feasible. Instead, as we detail in the paper, we have taken a considerate approach to using this data in a way that respects confidentiality and privacy, e.g. removing all usernames and paraphrasing all quotes. We hope that this is satisfying to the Editor and are happy to discuss the matter further if required.

5. Please remove all personal information, ensure that the data shared are in accordance with participant consent, and re-upload a fully anonymized data set.

Personal information is not included in the data set contained in the manuscript.

6. We note that [Supplementary File 2 - Blog with Proof Photos] includes an image of a participant in the study].

All photos and identifiable information, including images of panelists have now been removed from [Supplementary File 2 - Blog with Proof Photos]. This has now been replaced with a new attachment [S2 Supplementary Information. Promotional Blog].

Supporting Information file names have now been updated according to the guidelines.

The reference list is complete and correct. No cited papers have been retracted.

---

## [Editor Report · Decision Letter 1]

26 Nov 2025

5 years of bipolar disorder conversations on Reddit: Methods, key topics and future directions

PONE-D-25-19143R1

Dear Dr. Poh,

We’re pleased to inform you that your manuscript has been judged scientifically suitable for publication and will be formally accepted for publication once it meets all outstanding technical requirements.

Kind regards,

Souparno Mitra, M.D.

Academic Editor

PLOS ONE
---

## [Editor Report · Acceptance letter]

PONE-D-25-19143R1

PLOS One

Dear Dr. Poh,

I'm pleased to inform you that your manuscript has been deemed suitable for publication in PLOS One. Congratulations! Your manuscript is now being handed over to our production team.

Kind regards,

on behalf of

Dr. Souparno Mitra

Academic Editor

PLOS One